

# Examining the ability to track multiple moving targets as a function of postural stability: a comparison between team sports players and sedentary individuals

Teresa Zwierko[1], Piotr Lesiakowski[2], Beatriz Redondo[3] and Jesús Vera[3]

[1] Institute of Physical Culture Sciences, Laboratory of Kinesiology, Functional and Structural Human Research Center, University of Szczecin, Szczecin, Poland
[2] Department of Physical Education and Sport, Pomeranian Medical University, Szczecin, Poland
[3] CLARO (Clinical and Laboratory Applications of Research in Optometry) Research Group, Department of Optics, University of Granada, Granada, Spain

Corresponding author
Teresa Zwierko,
teresa.zwierko@usz.edu.pl

## ABSTRACT

**Background:** The ability to track multiple objects plays a key role in team ball sports actions. However, there is a lack of research focused on identifying multiple object tracking (MOT) performance under rapid, dynamic and ecologically valid conditions. Therefore, we aimed to assess the effects of manipulating postural stability on MOT performance.

**Methods:** Nineteen team sports players (soccer, basketball, handball) and sixteen sedentary individuals performed the MOT task under three levels of postural stability (high, medium, and low). For the MOT task, participants had to track three out of eight balls for 10 s, and the object speed was adjusted following a staircase procedure. For postural stability manipulation, participants performed three identical protocols (randomized order) of the MOT task while standing on an unstable platform, using the training module of the Biodex Balance System SD at levels 12 (high-stability), eight (medium-stability), and four (low-stability).

**Results:** We found that the ability to track moving targets is dependent on the balance stability conditions ($F_{2,66} = 8.7$, $p < 0.001$, $\eta^2 = 0.09$), with the disturbance of postural stability having a negative effect on MOT performance. Moreover, when compared to sedentary individuals, team sports players showed better MOT scores for the high-stability and the medium-stability conditions (corrected $p$-value = 0.008, Cohen's d = 0.96 and corrected $p$-value = 0.009, Cohen's d = 0.94; respectively) whereas no differences were observed for the more unstable conditions (low-stability) between-groups.

**Conclusions:** The ability to track moving targets is sensitive to the level of postural stability, with the disturbance of balance having a negative effect on MOT performance. Our results suggest that expertise in team sports training is transferred to non-specific sport domains, as shown by the better performance exhibited by team sports players in comparison to sedentary individuals. This study provides novel insights into the link between individual's ability to track multiple moving objects and postural control in team sports players and sedentary individuals.

# INTRODUCTION

In the highly dynamic and constantly changing scenario of team sports such as basketball, soccer or handball, athletes need to rapidly process a considerable amount of information in order to make appropriate decisions (*Ashford, Abraham & Poolton, 2021*; *Roca & Williams, 2016*). In this regard, the ability to track moving objects seems to be a crucial aspect of perceptual-cognitive function towards skilled performance in different sport disciplines (*Howard, Uttley & Andrews, 2018*; *Mackenzie et al., 2021*).

The multiple object tracking (MOT) test, which is based on the manipulation of spatiotemporal demands, has been developed to evaluate and enhance the ability to track targets within a dynamic environment where all objects are in constant motion (*Pylyshyn & Storm, 1988*). In team sports, there is scientific evidence showing that the speed of tracking multiple objects is positively associated with sport expertise in soccer (*Faubert, 2013*), rugby (*Harris et al., 2020*), and basketball (*Jin et al., 2020*; *Qiu et al., 2018*). Indeed, MOT performance has demonstrated to be associated with specific measures of game performance (assists, turnovers, assist-to-turnover ratio, steals) in professional basketball players (*Mangine et al., 2014*). Interestingly, a laboratory MOT training intervention improved passing decision-making in soccer players (*Romeas, Guldner & Faubert, 2016*) and enhanced processing speed and sustained attention in volleyball players (*Fleddermann, Heppe & Zentgraf, 2019*). Moreover, the ability to track moving targets seems to be associated with sport performance, and also, its improvement could have a positive impact on applied contexts.

In real game situations, a number of targets are in constant motion (*i.e.*, the opponent, teammates, the ball), and it usually occurs while players' moving. Indeed, team sports are characterized by the repeated combination of high-intensity actions such as sprints, jumps, accelerations, decelerations and multiple changes-of-direction, interspersed with brief low-intensity periods of running and standing (*Bishop & Girard, 2013*). To maintain the integrity of the sport-specific skills, team sports have a greater demand on coupling the athlete's perceptual-cognitive and motor subsystems (*Davids et al., 2001*; *Farrow & Abernethy, 2003*). This integrity between higher perceptual-cognitive function and the player's motor system has been confirmed by the analysis of effective motor behaviors in skilled athletes, as for example in soccer dribbling (*Fransen et al., 2017*), agility tasks performance (*Spiteri et al., 2018*), and defensive actions in soccer (*Roca et al., 2011*). In addition, dynamic balance is defined as the ability to control the postural stability during complex movements and challenging postural conditions (*e.g.*, during external mechanical perturbations) (*Paillard & Noé, 2015*). Regarding team sports, dynamic balance is considered as a functional prerequisite to perform complex motor skills such as ball control (*Paillard, 2017*) or agility tasks (*Stirling, Eke & Cain, 2018*). In this context, it seems appropriate to consider the bidirectional relationship between the motor and perceptual-cognitive functions in more realistic scenarios, namely when stability is

compromised. In our opinion, a lack of perception-movement coupling in research contexts is failing to replicate sport-specific situations, and thus, there is a knowledge gap that needs further exploration.

Based on the previously reported research gaps, the aim of the present study was to assess the impact of manipulating the level of postural stability on MOT performance in a sample of team sports players and sedentary individuals. It is expected that MOT performance would be positively associated with the level of stability since visual search performance has been linked to stability (*Marsh et al., 2010*). Also, it is hypothesized that athletes, when compared to non-athletes, would achieve greater MOT scores (*Howard, Uttley & Andrews, 2018*; *Qiu et al., 2018*) and have better dynamic postural control (*Reynard, Christe & Terrier, 2019*), resulting in a better MOT performance with different levels of stability.

## MATERIALS AND METHODS

### Participants

An a-priori sample size calculation was performed using G\*Power 3.1 (*Faul et al., 2007*), assuming an effect size of 0.25, alpha of 0.05, and power of 0.85. This analysis projected a minimum sample size of 32 participants (16 participants in each group) for the desired statistical power. A total of 35 males were included in this study, 19 professional and semiprofessional team sports players (soccer: $n = 6$; basketball: $n = 7$; and handball: $n = 6$) and 16 university students, who did not regularly practice physical activity and declared a sedentary lifestyle. In our study the sedentary individual was defined as a physical inactivity person (physical activity of less than 30 min per day) who spent his/her time mostly on screen-based leisure activities (*e.g.*, television watching, internet use, and other forms of screen-based entertainment) and/or screen-based work activities (*Panahi & Tremblay, 2018*) (see Table 1 for a description of the experimental sample). All participants had no history of major lower limb injury and were free of any visual deficit.

All participants were informed about the testing procedure, and signed a written informed consent. This study was approved by the University of Granada's Institutional Review Board (IRB approval: 1180/CEIH/2020).

### Postural stability assessment

Participants were tested individually, and all assessments were conducted in the same room under constant environmental conditions (no variation in temperature and lighting, a lack of background noises or other environmental cues). Initially, the bilateral static and dynamic postural stability tests were carried out by using the Biodex Balance System SD (Biodex Medical Systems Inc, Shirley, NY, USA). Postural stability tests were performed on static (rigid surface setting) and dynamic platforms (multiaxial platform with 12 levels of instability, maximum tilt of 20 degrees). Test duration for each of the two balance tasks was 80 s (three trials of 20 s each, with a rest interval of 10 s between each). The dynamic postural stability test was performed with platform stability on levels 8 to 4. For all trials, participants were tested barefoot. During testing, participants looked straight ahead to a reference point with their arms folded along their chest. The overall stability index

**Table 1 Descriptive (mean ± standard deviation) characteristics of the experimental sample, and its statistical comparison between groups.**

|  | Team sports players (*n* =19) | Sedentary individuals (*n* = 16) | *p*-value |
| --- | --- | --- | --- |
| Age (years) | 20.7 ± 2.6 | 19.7 ± 2.0 | 0.222 |
| Height (cm) | 188.1 ± 8.0 | 183.9 ± 6.2 | 0.099 |
| Weight (Kg) | 82.2 ± 12.0 | 78.3 ± 9.5 | 0.301 |

(OSI) (°), the anterior-posterior stability index (APSI) (°), and the medial-lateral stability index (MLSI) (°) were determined. Stability indexes represent fluctuations around a zero point, which was established prior to testing and when the platform was stable. The average value of three trials was considered for further analyses (*Arnold & Schmitz, 1998*). Previous studies have reported the level of intra-tester reliability for the Biodex Balance System (*Cachupe et al., 2001*; *Schmitz & Arnold, 1998*), and the ICC values obtained in the current study procedure for OSI, APSI, and MLSI were 0.89, 0.83, and 0.61 during static condition and were 0.75, 0.73 and 0.68 during dynamic condition, respectively. Higher scores of stability index indicate poorer postural stability.

## Multiple object tracking (MOT)

Following previously described procedure for the MOT test by *Vera et al. (2022)*, eight identical black balls (diameter 2.06°) were projected on a 65 cm white square background with a luminance of 107 cd/m2, which subtended a visual angle of 36°, using a 55-inches television monitor (UE55NU7172; Samsung, Seoul, Korea) placed at 1 m. For each trial, three of these balls were randomly highlighted in green for 2 s before returning to the baseline black color. The participant was instructed to track these three balls for 10 s. The examiner did not give any specific instruction about how performing the task. As in real-world contexts, eye movements were allowed so that they could use a gaze strategy to better perform the task (*Fehd & Seiffert, 2008*, *2010*). Participants had no previous experience with the MOT task. All balls moved in random linear trajectory with a constant speed. The balls only deviated from a linear path when they crossing and bouncing each other or the walls. After 10 s, all the balls were stopped in place and a number (from 1 to 8) was assigned to each one. At the end of each trial, the participant reported the final perceived position of three balls that were originally highlighted based on their location in the display (*Fehd & Seiffert, 2008*). A graphical illustration of the MOT testing procedure is presented on Fig. 1C. In this study, the speed of the balls was adjusted with a 1-up 1-down staircase procedure (*Levitt, 1971*), *i.e.*, increasing the speed if all three balls were identified correctly or decreasing the speed if at least one ball was identified incorrectly. The initial speed of the balls was set at 26.3 cm/s. After each correct or incorrect trial the speed was increased or decreased by 0.05 log, respectively. The staircase stopped after six effective reversals had occurred. A mean of the speeds of the last four reversals was taken as the measure of MOT performance.

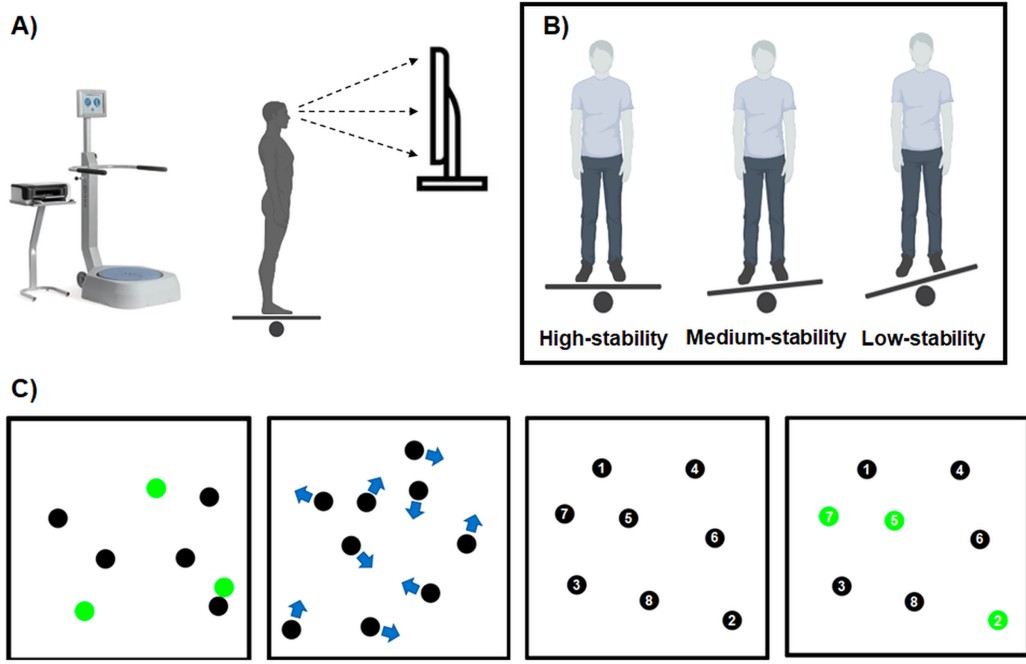

**Figure 1 A graphical illustration of the testing procedure.** (A) Starting position where the participant was standing on an unstable platform working at the training module of the Biodex Balance System SD placed 1 m in front of the television monitor; (B) three levels of platform stability: high (level 12), medium (level 8) and low (level 4); (C) four stages of the MOT task, *i.e.*, presentation stage where three out of eight targets (balls) were temporarily (2 s) highlighted on green color; movement stage where the targets were at the same color (black) and all moved for 10 s crossing and bouncing each other; identification stage where the targets were frozen and marked with numbers, and the participant had to identify by giving three numbers of balls originally highlighted in the presentation stage; feedback stage where the participant was given information of the correct targets.

## Procedure

To complete the MOT task, each participant performed three testing conditions (three levels of stability) in a randomized manner with a rest interval of 10 min between two consecutive conditions. During the execution of the MOT task, participants tried to keep balance on an unstable platform working at the training module of the Biodex Balance System SD. Each testing session was different with levels of platform stability, (*i.e.*, level 12 (high stability with maximum platform tilt of 1.7°), level 8 (medium stability with maximum platform tilt of 8.4°) and level 4 (low stability with maximum platform tilt of 15.0°)). An experienced examiner gave standardized instructions and monitored the testing procedure. All assessments had a standardized familiarization protocol, which included two MOT trials using the initial speed (26.3 cm/s). Figure 1 depicts a graphical illustration of the testing procedure.

## Statistical analyses

Descriptive data are presented as means and standard deviations. The normal distribution of the data (Shapiro-Wilk test) and the homogeneity of variances (Levene's test) were confirmed ($p > 0.05$). In order to determine the possible differences between team sports players and sedentary individuals for OSI, APSI, and MLSI, three separate t-tests for

**Table 2 Descriptive and statistical values for static and dynamic OSI, APSI, and MLSI in the groups of team sports players and sedentary individuals.**

| Postural balance | Stability index | Team sports players | Sedentary individuals | p-value (Cohen's d) |
|---|---|---|---|---|
| Static | OSI (°) | 0.311 ± 0.221 | 0.369 ± 0.260 | 0.479 (0.243) |
| | APSI (°) | 0.216 ± 0.201 | 0.275 ± 0.198 | 0.388 (0.297) |
| | MLSI (°) | 0.153 ± 0.077 | 0.163 ± 0.182 | 0.831 (0.073) |
| Dynamic | OSI (°) | 0.884 ± 0.257 | 0.956 ± 0.346 | 0.485 (0.240) |
| | APSI (°) | 0.658 ± 0.295 | 0.669 ± 0.336 | 0.920 (0.035) |
| | MLSI (°) | 0.526 ± 0.268 | 0.569 ± 0.265 | 0.642 (0.159) |

**Note:**
OSI, overall stability index; APSI, anterior-posterior stability index; MLSI, medial-lateral stability index.

independent samples were carried out. For the main analysis, a mixed ANOVA with "stability level" as the only within-participants factor, and "group" as the only between-participants factor, was performed for MOT score. The possible associations between stability indexes (OSI, APSI, and MLSI) in static and dynamic conditions with MOT scores were assessed by separate linear regression analyses. A $p$-value of 0.05 was considered to determine statistical significance, and the magnitude of the differences (effect sizes) were reported using the Cohen's d (d′) and eta squared ($\eta^2$) for t- and F-tests, respectively. The criteria for interpreting the magnitude of the effect sizes were: trivial (<0.2), small (0.2–0.6), moderate (0.6–1.2), large (1.2–2.0) and extremely large (>2.0) for Cohen's d (*Hopkins et al., 2009*) and small (0.01), medium (0.06), and large (0.14) for eta squared (*Cohen, 1988*). *Post-hoc* comparisons were corrected by the Holm-Bonferroni procedure, and the JASP statistical package (version 16.1) was used for all analyses.

## RESULTS

Descriptive and statistical values for static and dynamic OSI, APSI, and MLSI in the groups of team sports players and sedentary individuals are shown in Table 2.

In the static postural balance task, team sports players did not statistically differ from sedentary individuals in terms of stability indexes ($p$-values > 0.05, Cohen's ds ranging from 0.073 to 0.297). Similarly, in the dynamic postural balance task, there were no statistically significant differences between both experimental groups ($p$-values > 0.05, Cohen's ds ranging from 0.035 to 0.240).

For the analysis of MOT performance, the main effects of "stability level" ($F_{2,66} = 8.7$, $p < 0.001$, $\eta^2 = 0.09$) and "group" ($F_{1,33} = 10.9$, $p = 0.002$, $\eta^2 = 0.15$) reached statistical significance, but the interaction "stability level" × "group" was not statistically significant ($F_{2,66} = 1.9$, $p = 0.678$, $\eta^2 = 0.01$) (Fig. 2). Regarding stability level, greater MOT scores were found for the high-stability in comparison to the medium-stability (corrected $p$value < 0.001, Cohen's d = 0.67) and low-stability (corrected $p$-value = 0.005, Cohen's d = 0.54) conditions. However, the comparison between the medium-stability and low-stability conditions did not reveal statistically significant differences (corrected $p$value = 0.444, Cohen's d = 0.13). Statistically significant *post-hoc* comparisons between both experimental groups for each stability level are depicted in Fig. 2.

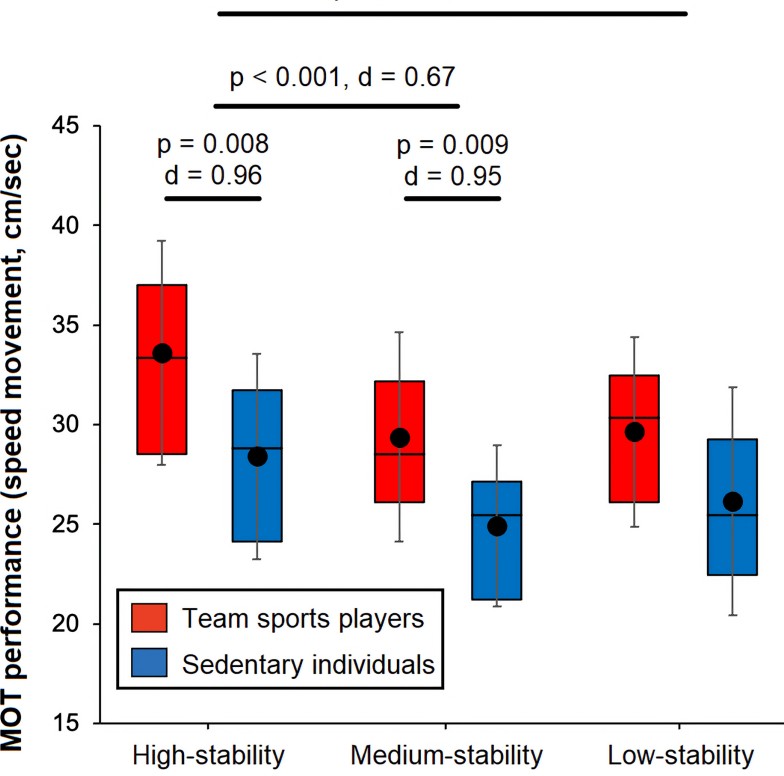

**Figure 2 Boxplot of the effect of stability conditions on multiple objects tracking performance in a group of team sports players (in red) and sedentary individuals (in blue).** Statistically significant differences are depicted in the figure (Holm-Bonferroni corrected *p*-value < 0.05), and the magnitude of the differences are reported by Cohen's d. The box plots represent 75th, 50th and 25th percentiles. Horizontal lines and circles into the box represent median and mean values, respectively. The whiskers show the standard deviation.

The analysis of the association between stability indices and changes in MOT performance across conditions showed that either static and dynamic postural balance were not correlated with MOT performance. However, there were positive correlations between sports experience and MOT scores in the high-stability (r = 0.414, p = 0.013) and medium-stability (r = 0.365, p = 0.031) conditions (Fig. 3).

## DISCUSSION

We examined the effects of manipulating postural stability on the ability to track moving objects in team sports players and sedentary individuals. Our main findings are that, when compared to sedentary individuals, team sports players showed better MOT scores for the high-stability and medium-stability conditions whereas no between-groups differences were reached in the more unstable conditions (low-stability). Also, the manipulation of postural stability showed to have an effect on MOT performance, showing that the ability to track moving targets is dependent on the stability conditions.

Expertise from the sport domain characterized by dynamically changing, high-paced and unpredictable scenario may transfer to a more general perceptual-cognitive domain

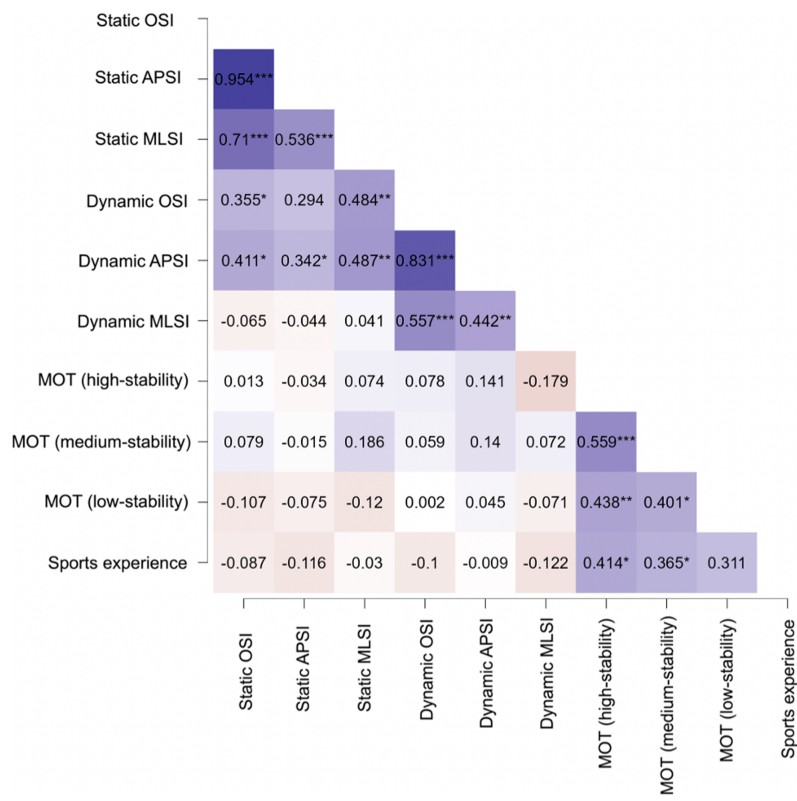

**Figure 3 Heat map showing separate linear regression analyses between the different variables assessed in this study.** $^*p < 0.05$, $^{**}$ $p < 0.01$, $^{***}$ $p < 0.001$

(*i.e.*, MOT) (*Faubert, 2013*; *Harris et al., 2020*; *Howard, Uttley & Andrews, 2018*; *Jin et al., 2020*; *Qiu et al., 2018*). In a study using intracranial electrophysiological recordings (visual evoked potentials, VEP) in athletes practicing volleyball for 2 years, *Zwierko et al. (2014)* observed a reduction of VEP components latencies (P100 and N75), suggesting that the effects of regular sport training cause improvements in the sensory stage of information processing. Another analysis of event related potentials (P300 latency and P300 amplitude) showed a link between sports experience and cerebral cortical activity during a perceptual task, with skilled cricket batsmen having superior decision-making ability in comparison to less-skilled players (*Taliep et al., 2008*). Specifically, *Qiu et al. (2019)* reported that the neural efficiency of better MOT performance in team sport athletes is associated with bidirectional reductions in cortical activation and deactivation. In fact, these authors found that during the execution of a MOT task, athletes demonstrated less activation in attention-related brain areas and less deactivation in the medial superior frontal gyrus in comparison to non-athletes. Taken together, the results of this study corroborate that team sports players have a greater ability to track moving targets than individuals who do not regularly practice physical activity.

Our findings suggest that the advantage of athletes over non-athletes in MOT scores may result mainly from perceptual-cognitive expertise and enhanced ability to perception-action coupling, rather than a better postural control. Somewhat surprisingly,

the initial scores of dynamic overall stability index indicated non-statistically significant differences between groups, with the magnitude of the differences being negligible to small (Cohen's d s ≤ 0.240). Although, it is widely accepted that postural performance is improved after regular sport activity (*Reynard, Christe & Terrier, 2019*), it is also known that in experienced athletes the postural balance adaptation is very specific to the context of the sport practice, therefore an effect of its transfer to non-specific contexts is modest or inexistent (*Paillard, 2017*). Moreover, morphological parameters of athletes, such is a higher body height, may also have some influence on the postural stability test results. Indeed, body height is recognized as the anthropometric variable with greater influence on postural balance (*Alonso et al., 2012*), which may partially explain the current results ($p$value = 0.099 for the height differences between groups).

Despite the differences in MOT performance between team sports players and sedentary individuals, the changes in MOT scores under increasing postural instability was similar in both experimental groups. In other words, the ability to track moving objects was modulated as a function of postural stability regardless of sport experience. Given the complexity of the task used (*i.e.*, MOT in unstable conditions) in this investigation, the integration of multiple sensory inputs and the coordination of multiple motor outputs is required. The results obtained may be explained by the uncoupling of the perceptual-cognitive and motor systems as result of the disturbance caused by compromising postural balance (*Vidal & Lacquaniti, 2021*). On the other hand in dual-task conditions, it has been indicated that cognitive and motor tasks may have a shared attentional resource and a redistribution of the attentional capacity to each task is plausible (*Long & Ma-Wyatt, 2014*). In challenging spatiotemporal conditions, attention narrows to goal-directed orientation (*i.e.*, objects' tracking), limiting the cognitive/motor processing linked to keep balance on an unstable platform (*Abernethy, 1993*). This competition for attention may negatively affect the motor control system, resulting in a dysfunction of the perceptual-cognitive and motor flow integrity (*Tenenbaum & Land, 2009*). Of note, the cognition-action interaction in the domain of visual attention involves arousal processes (*Davranche & Audiffren, 2004*), but also, inhibitory control processes play a role in this activity (*Tiego et al., 2018*). Recently, *Park, Ahn & Zhang (2021)* examined the impact of performing physical effort (handgrip exertion) at two intensity levels on visual search. They found a faster behavioral performance with physical effort due to the arousing effects of handgrip exertion, however, the most physically demanding condition caused a heightened interference from the singleton distractor and impaired cognitive performance as consequence of the reduced inhibitory control. Moreover, perceptual-cognitive skills seem to be highly dependent on the specific context of assessment, as corroborated by the manipulation of the stability conditions in the current study.

It is also plausible to hypothesize that changes in MOT performance results during the increasing instability of the platform were caused by oculomotor system disturbances. During the execution of the MOT task, the observer is required to maintain its fixation, specifically when the center-looking strategy (attending to all the targets as a group) is used (*Fehd & Seiffert, 2008*), which consequently causes the inhibition of eye movements (*Howe*

*et al., 2009*). On the contrary, postural balance in dynamic conditions is controlled by the use of saccadic eye movements or smooth pursuit movements which, in contrary to fixation, attenuate postural sway (*Rodrigues et al., 2015*; *Zwierko, Lesiakowski & Zwierko, 2020*). The issue of oculomotor coordination when performing tasks with concomitant demands of different nature is worth further investigated. Future research should try to investigate the eye movement strategies that lead to successful tracking of moving objects in unstable conditions.

The current results provide novel insights into the relationship between the ability to track multiple moving targets and the level of postural stability. However this study is not exempt of limitations and they must be acknowledged. First, our experimental sample was formed by athletes from three sport disciplines (*i.e.*, soccer, basketball, and handball). There is scientific evidence that the ability of attentional control in MOT tasks varies across sport disciplines (*Harris et al., 2020*), and even across representatives of the same sport discipline as an effect of playing position on the court (*Mangine et al., 2014*; *Martín et al., 2017*). Although, all analysed team sports in this study belong to the category of open skill sports, it is possible that skill expertise of particular sport and position of players may affect our study results. It was found that experts in open-skills sport performed visual tracking tasks better than novice and intermediate players (*Qiu et al., 2018*). Specifically, the acquisition of specific skill expertise collected from training process is accompanied by players' structural and functional reorganization of the brain (*Nakazawa, 2022*; *Gao et al., 2019*). A study using deterministic tractography to identify streamlines connecting cortical and subcortical brain regions of elite open-skills sport athletes indicated that skill expertise allows to optimize the visuomotor processing (brain's expressways), causing a better integration of information about externally-paced environments and motor control (*Zhu et al., 2019*). Second, previous studies have shown a gender-effect on the ability to track multiple objects (*Roudaia & Faubert, 2017*) and thus, the level of association between the MOT task and dynamic postural stability could differ between men and women. The experimental sample was limited to male young adults, and the external validity of these findings to other populations should be addressed in future investigations. Third, while the current findings support the potential utility of including MOT for team sport training, further studies examining the relationship between MOT performance in ecological contexts (*e.g.*, under dynamic conditions) and game-related performance are needed. Fourth, in our study MOT performance was obtained as the speed at which participants were able to track three targets following a staircase procedure. However, it would be interesting to evaluate the number of balls that team sports players and sedentary individuals are able to track following the procedures described by *Alvarez & Franconeri (2007)*.

## CONCLUSIONS

Our data exhibit that team sports players have a better ability to track multiple moving targets under different levels of postural stability than sedentary individuals. The ability to track moving targets is sensitive to the postural stability level, with the disturbance of postural stability having a negative effect on MOT performance. These findings provide

novel insights into the link between individual's ability to track multiple moving objects and postural control in team sports players and sedentary individuals.

### Funding
The authors received no funding for this work.

### Competing Interests
Beatriz Redondo and Jesús Vera are Academic Editors for PeerJ.

### Author Contributions
- Teresa Zwierko conceived and designed the experiments, performed the experiments, prepared figures and/or tables, authored or reviewed drafts of the article, and approved the final draft.
- Piotr Lesiakowski conceived and designed the experiments, performed the experiments, authored or reviewed drafts of the article, and approved the final draft.
- Beatriz Redondo conceived and designed the experiments, analyzed the data, authored or reviewed drafts of the article, and approved the final draft.
- Jesús Vera conceived and designed the experiments, analyzed the data, prepared figures and/or tables, authored or reviewed drafts of the article, and approved the final draft.

### Human Ethics
The following information was supplied relating to ethical approvals (*i.e.*, approving body and any reference numbers):

The Institutional Ethics Committee reviewed and authorized the research protocol (IRB approval: 1180/CEIH/2020).

### Data Availability
The raw measurements are available in the Supplemental File.

### Supplemental Information
Supplemental information for this article can be found online at http://dx.doi.org/10.7717/peerj.13964#supplemental-information.

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
