# Peer review of "Examining the ability to track multiple moving targets as a function of postural stability: a comparison between team sports players and sedentary individuals"

_PeerJ, doi:10.7717/peerj.13964_

## Round 0.1 · original submission · Minor Revisions

Dear Authors

Your submission has been reviewed by two expects in the field of study. The comments of the reviewers are included at the bottom of this letter. We invite you to submit a revised version of the manuscript that addresses the points raised by the reviewers.

We look forward to receiving your revised manuscript.

Best regards

Yung-Sheng Chen, PhD
Academic Editor

Reviewer 1 ·

Basic reporting

This manuscript is well-written with a good article structure, and clear figures and tables. Overall, this study investigated the effect of manipulating postural stability on multiple object tracking (MOT) performance in two different groups. The main finding was that compared with sedentary individual, team sports players have better MOT ability in different manipulating postural stability, and the interference of manipulating postural stability has a significant effect on MOT performance. MOT ability has a negative impact and provides new insights into postural control in sedentary individual and team sports players.

Experimental design

1. Did this study allow eye movements during MOT? What's the difference with the non-movements? Did it affect the results? Furthmore, Pylyshyn, Z. W., & Storm, R. W. (Spat Vis. 1988;3(3):179-97.) have shown that the analysis authors used may not the most appropriate for this situation. Please explain why author still adopted this method.
2. In this study, three balls were selected at the end of the MOT? What is the difference from the previous study by Alvarez, G. A., & Franconeri, S.L.( J Vis. 2007 Oct 30;7(13):14.1-10.) that the maximum number of balls can be set to eight? Also, are there different numbers of objects that can be tracked between professional players and non-professional players? Please clarify.
3. Regarding the data of the Biodex balance system, was it the average of the three trials or the highest value? It is recommended to cite additional sources of literature. In addition, please add intraclass correlation coefficient (ICC) and explain what is meant by a constant environment. For your reference, as, Sherief, A. et al., (J Musculoskelet Neuronal Interact. 2021; 21(3): 343–350.)
4. Please clarify the definition of a sedentary individuals and cite reference. In addition, were the subjects recruited for this study only male in the sedentary individuals? If so, why? Although the limitations are mentioned. My concerns is that the bias caused by sports? please clarify.
5. Different sports and different positions of players also affect the outcomes. It is recommended to be mentioned in the discussion, since many literatures has reported this issues. As, Qiu, F. et al., PeerJ. 2018 Sep 28;6:e5732, and Martin, A. et al., Front Psychol. 2017 Sep 8;8:1494, and Alvarez, G. A. et al., J Vis. 2007 Oct 30;7(13):14.1-10.)
6. Whether the subject has experience with MOT, my concern that it may exist a learning effect, if not, please clarify.

Validity of the findings

statistically sound, report good

Additional comments

Introduction. The last paragraph is recommended to write a simple description, since the same text has been repeated in the material part. Please revise.

·

Basic reporting

A well-written paper that clearly articulates the literature.

Experimental design

Experimental design is appropriate and the level of detail and information for replication is sufficient.

Validity of the findings

The results are somewhat important and provide insight into the topic area. Although the external validity is limited due to the laboratory-based nature of the measures, it still provides some important findings that set the foundation for future, more applied investigations. However, some of the conclusions are not entirely supported by the results.

Additional comments

I thank the authors for presenting such a clear and well-designed manuscript. It was very clear in its writing, with only minor grammatical errors. The methods were clearly explained and were appropriately designed to answer the research question. I only had a few minor issues with some of the main conclusions and discussion which can be easily remedied with minor changes.

---

## Round 0.2 · accepted · Accept

Dear Authors

I'm pleased to inform you that the reviewers fully agreed the acceptance of your manuscript for publication. Congratulations.

Best Regards

Yung-Sheng Chen, PhD
Academic Editor

Reviewer 1 ·

Basic reporting

clear

Experimental design

well-defined and performed

Validity of the findings

well stated

Additional comments

Authors have addressed my concerns.

·

Basic reporting

The manuscript is well written with very clear diagrams.

Experimental design

The experimental design and write-up are well thought out and presented.

Validity of the findings

The changes that have been made in relation to the discussion of the findings and the main conclusions are both satisfactory and pleasant to see.

Additional comments

Thank you for providing clear responses to the original review.